# Evaluation of an Alternative Screening Method for Gestational Diabetes Diagnosis During the COVID-19 Pandemic (DIABECOVID STUDY): An Observational Cohort Study

**DOI:** 10.3390/diagnostics15020189

**Published:** 2025-01-15

**Authors:** Alba Casellas, Cristina Martínez, Judit Amigó, Roser Ferrer, Laia Martí, Carme Merced, Maria Carmen Medina, Istria Molinero, Marta Calveiro, Anna Maroto, Ester del Barco, Elena Carreras, Maria Goya

**Affiliations:** 1Maternal-Foetal Medicine Unit, Department of Obstetrics and Gynecology, Hospital Universitari Vall d’Hebron, Universitat Autònoma de Barcelona, 08036 Barcelona, Spainmaria.goya@vallhebron.cat (M.G.); 2Sexual and Reproductive Health Services, Catalan Institute of Health, Universitat de Barcelona (UB), 08007 Barcelona, Spain; 3Department of Endocrinology, Hospital Universitari Vall d’Hebron, Universitat Autònoma de Barcelona, 08036 Barcelona, Spain; 4Department of Clinical Biochemistry, Hospital Universitari Vall d’Hebron, Universitat Autònoma de Barcelona, 08036 Barcelona, Spain; 5Department of Obstetrics and Gynecology, Hospital Universitari Parc Taulí, 08208 Sabadell, Spain; 6Department of Obstetrics and Gynecology, Consorci Hospitalari de Vic, 08500 Barcelona, Spain; 7Department of Obstetrics and Gynecology, Hospital de la Creu i Sant Pau, 08025 Barcelona, Spain; 8Department of Obstetrics and Gynecology, Hospital De Igualada, 08700 Barcelona, Spain; 9Atenció a la Salut Sexual i Reproductiva (ASSIR) Muntanya, 08035 Barcelona, Spain; 10Department of Obstetrics and Gynecology, Hospital Josep Trueta, 17007 Girona, Spain

**Keywords:** gestational diabetes, COVID-19, O’Sullivan, OGTT, glucose level, glycated hemoglobin, screening, macrosomia, large for gestational age (LGA), neonatal hypoglycemia

## Abstract

**Background:** To evaluate the impact of applying alternative diagnostic criteria for gestational diabetes mellitus (GDM) during the COVID-19 pandemic on GDM prevalence, obstetrical and perinatal outcomes, and costs, as compared to the standard diagnostic method. **Methods:** A cohort of pregnant individuals undergoing GMD screening with the alternative GDM method, which uses plasma glucose (fasting or non-fasting) and HbA1c, was compared with a cohort of pregnant individuals undergoing the standard GDM screening method. Both cohorts were obtained from six hospitals across Catalonia, Spain, from April 2020 to April 2022. The primary outcome was large for gestational age rate at birth. The secondary outcomes were composite adverse outcomes, including pregnancy complications, delivery complications, and neonatal complications. The cost differences between screening methods were also evaluated. A similar analysis was performed in the subgroup diagnosed with GDM. **Results:** Data were collected from 1543 pregnant individuals in the standard screening group and 2197 in the alternative screening group. The standard screening group had a higher GDM diagnostic rate than the alternative screening group (10.8% vs. 6.9%, respectively; *p* < 0.0001). The primary outcome (large for gestational age rate) was similar between groups: 200/1543 (13.0%) vs. 303/2197 (13.8%). The adjusted OR for this outcome was 1.74 (95% CI: 0.74–4.10). An adjusted analysis showed no differences between groups in the composite adverse outcomes for pregnancy complications (OR: 1.11; 95% CI: 0.91–1.36), delivery complications (OR: 0.95; 95% CI: 0.75–1.19), and neonatal complications (OR: 1.28; 95% CI: 0.94–1.75). Among individuals diagnosed with GDM, the large for gestational age rate was similar between groups: 13/166 (7.8%) vs. 15/151 (9.9%). The OR adjusted for this outcome was 1.24 (95% CI: 0.51–3.09). An adjusted analysis showed no differences in the composite adverse outcomes for pregnancy complications (OR: 1.57; 95% CI: 0.84–2.98), delivery complications (OR: 1.21; 95% CI: 0.63–2.35), and neonatal complications (OR: 1.35; 95% CI: 0.61–3.04). The mean cost (which included expenses for consumables, equipment, and personnel) of the alternative screening method was 46.0 euros (22.3 SD), as compared to 85.6 euros (67.5 SD) for the standard screening method. **Conclusions:** In this Spanish population during the COVID-19 pandemic, GDM prevalence was lower in the alternative screening group than in the standard screening group. After adjusting for GDM risk factors, outcomes related to obstetrics, delivery, and neonatal complications were comparable between both groups. Finally, the alternative screening method was cheaper than the standard screening method.

## 1. Introduction

The diagnostic challenges of gestational diabetes mellitus (GDM) have sparked extensive debates among experts. The primary areas of debate include selecting between universal and selective screening, two-step versus one-step diagnostic approaches, in particular, the oral glucose tolerance test (OGTT), selecting the most suitable diagnostic test (e.g., 100 g or 75 g OGTT), and establishing appropriate cut-off values for diagnosis [1,2]. To make matters worse, the COVID-19 pandemic has added an unexpected layer of complexity to this ongoing dilemma.

During the COVID-19 pandemic, it became apparent that pregnant individuals faced significant risks when attending healthcare sites for GDM screening following the usual two-step approach, including the 50 g O’Sullivan test (OGTT) with 1 h glucose measurement and the 100 g OGTT with 3 h glucose measurement. Consequently, several international societies, including the Spanish Group of Diabetes and Pregnancy (GEDE) and the Catalan Department of Health, developed an alternative approach to the standard GDM screening method [3,4,5,6,7]. This alternative method streamlines the screening process through a one-step process that includes plasma glucose (fasting or non-fasting) and hemoglobin A1c (HbA1c). This approach minimizes prolonged hospital stays [3,6,8,9,10,11]. Recently, several researchers have evaluated the consequences of implementing these alternative diagnostic criteria for gestational diabetes by comparison to a historical dataset. The suggested modifications to the diagnostic criteria for GDM in a number of countries have resulted in different GDM prevalences [12]. Some of these findings indicate an increased number of cases with undiagnosed GDM, of up to 30–50% [13,14]. On the other hand, when using the alternative diagnostic criteria, a Spanish cohort showed a GDM prevalence during the COVID-19 pandemic similar to that found in 2019, when the standard diagnostic criteria were being used [15].

In response to the unique circumstances during the COVID-19 pandemic, different regions and healthcare domains decided to modify their GDM diagnostic criteria. In April 2020, some Catalan regions adjusted their standard criteria to adapt to the pandemic circumstances. Since then, the criteria for GDM screening adapted to COVID-19 circumstances based on Spanish recommendations have been used in those regions during the pandemic and post-pandemic, although currently most regions have returned to the standard screening method. By contrast, other regions did not see a reason to change their standard GDM screening method.

The primary aim of this study was to assess the impact of following alternative diagnostic criteria for gestational diabetes mellitus (GDM) during the COVID-19 pandemic on GDM prevalence, obstetrical and perinatal outcomes, and cost as compared to the standard diagnostic criteria.

## 2. Material and Methods

An observational cohort study was performed to investigate the impact of replacing the standard GDM screening method with an alternative screening method to adapt to the circumstances of the COVID-19 pandemic. This study was approved by the Vall d’Hebron Barcelona Hospital Campus Ethics Committee (VHEC) PR(AMI)381/2020 in October 2020. This study was also approved by the Ethics Committee of each site. The study period ranged from April 2020 to April 2022. When the COVID-19 pandemic started, an alternative Spanish method for GDM screening was developed and published in May 2020. This alternative method was implemented and performed from May 2020 to April 2022. The retrospective period of the study included a cohort of patients undergoing the standard GDM screening from May 2020 to November 2020. Patient consent was waived for this cohort due to its retrospective nature. Informed consent was obtained from all individuals included in the prospective period of the study (November 2020 to April 2022). This work complied with the guidelines for human studies and was conducted ethically following the World Medical Association Declaration of Helsinki.

A cohort of pregnant individuals undergoing an alternative GMD screening (based on plasma glucose and HbA1c, fasting or non-fasting) due to the COVID-19 pandemic was compared with a cohort of pregnant individuals undergoing the standard GMD screening. Participants for both cohorts were obtained from 6 hospitals across Catalonia, Spain, from April 2020 to April 2022. The participating hospitals were Hospital Vall d’Hebron, Hospital Universitari Parc Tauli, Consorci Hospitalari de Vic, Hospital de Igualada, Hospital de Sant Pau, and Hospital Josep Trueta. Each participating hospital, following their internal protocols for the prevention of in-hospital COVID transmission, decided to either maintain the standard GDM screening protocol or switch to an alternative method proposed by different medical societies [11]. Thus, two cohorts were obtained: a group undergoing the standard GMD screening and a group undergoing the alternative GMD screening.

The standard GDM screening consisted of a two-step format, and tests were conducted in the first, second, and third trimesters [16]. In the first trimester, screening was carried out following the Catalan regional guidelines, provided risk factors were detected. These risk factors included age > 35 years, obesity (BMI ≥ 30 kg/m^2^ or 27.5 kg/m^2^ in Asian Americans), a first-degree family history of diabetes, and specific ethnic backgrounds (African Americans, Latino Americans, Pacific Islanders, Asian Americans). The test performed during the first trimester consisted of fasting glucose. If glucose levels exceeded 92 mg/dL, a 1 h O’Sullivan test (OGTT) involving the ingestion of 50 g of glucose was administered. If glucose levels after one hour were ≥140 mg/dL, a 100 g OGTT was conducted. The diagnostic cut-off levels for fasting glucose in the standard GDM screening were ≥105 mg/dL for fasting, ≥190 mg/dL for 1 h, ≥165 mg/dL for 2 h, and ≥145 mg/dL for 3 h glucose measurements (at least two of these thresholds were required for a diagnosis). Universal GDM screening was performed in the second trimester, between 24 and 28 gestational weeks. Third-trimester screening was limited to cases with large-for-gestational-age (LGA) fetuses, polyhydramnios, or when no screening had been conducted in the earlier trimesters.

The alternative screening method is also conducted in the first trimester only if risk factors are detected, in the second trimester for all patients, and in the third trimester only in some cases. Third-trimester screening is performed in cases with LGA fetuses or polyhydramnios, or if GDM screening is not conducted during the first or second trimester. The alternative screening method involves measuring both HbA1c and plasma glucose levels (preferably non-fasting due to feasibility or if fasting is not possible). The established cut-off values for GDM diagnosis in the first trimester were as follows: HbA1c ≥ 5.9%, fasting PG ≥ 100 mg/dL, and non-fasting PG ≥ 165–199 mg/dL. The established cut-off values for GDM diagnosis in the second trimester were HbA1c ≥ 5.7%, fasting PG ≥ 95 mg/dL, and non-fasting PG ≥ 165–199 mg/dL [11].

The inclusion criteria were being between 18 and 50 years old and the availability of perinatal outcomes. Participants with multiple gestation or pre-existing type-1 or type-2 diabetes were excluded.

All pregnant individuals included adhered to the established prenatal care protocol of each site, which was based on national guidelines [17,18]. The group diagnosed with GDM underwent a specific diet and glucose self-monitoring. Insulin was added as an additional treatment according to recommendations of Grupo Español de Diabetes y Embarazo (GEDE) [19].

Demographic, pregnancy, delivery, and neonatal data were collected using an electronic form by RedCap, a web-based record specifically designed for this study.

Demographic data, including maternal age, ethnicity, smoking status, family history of DM, previous GDM, and parity, were collected.

Pregnancy outcomes were collected. Gestational hypertension was classified as the worsening of chronic hypertension, new-onset gestational hypertension, and preeclampsia, up to 3 months postpartum [20]; polyhydramnios was defined as an amniotic fluid index ≥ 25 cm [21]; LGA was defined as a birth weight above the 90th percentile according to customized growth curves [22]; small for gestational age (SGA) was defined as a birth weight below the 10th percentile according to customized growth curves [3]; gestational loss included miscarriage, antepartum death, and intrapartum death; and gestational age at delivery was defined in complete weeks based on the date of the last menstrual period (LMP) or using the earliest ultrasound if there was a discrepancy with the LMP [3].

Delivery data were also collected. Obstetric trauma was defined as shoulder dystocia, clavicle fracture, or brachial plexus injury; the type of delivery was defined as vaginal delivery or caesarean section; maternal stay in the hospital was defined as the number of days after delivery at hospital for the mother; and the macrosomia rate was defined as a birth weight > 4000 g.

Neonatal outcomes were also recorded. Congenital anomalies were classified by EUROCAT code references; perinatal death was defined as fetal death from week 28 up to 7 days postpartum; neonatal hypoglycemia was defined as plasma glucose levels < 45 mg/dL within the first 24 h of life and <50.5 mg/dL afterwards; neonatal hypocalcemia was defined as calcium < 7 mg/dL; neonatal hyperbilirubinemia was defined as hyperbilirubinemia treated with phototherapy; neonatal polycythemia was defined as a hematocrit of a peripheral venous sample > 65%; respiratory distress syndrome was defined as progressive respiratory failure shortly after birth, along with certain features in a chest X-ray; excluding other causes; and hypertrophic cardiomyopathy was defined as a left ventricular wall thickness ≥ 15 mm by echocardiography.

Cost data were also taken into consideration. Expenses for lab tests (50 g and 100 g glucose bottles, blood glucose test, and glycated hemoglobin test) and personnel for conventional analysis were collected and compared between groups.

The primary outcome for both groups was defined as a large for gestational age (LGA) rate at birth. Secondary outcomes were composite adverse outcomes including pregnancy complications (macrosomia at birth, LGA at birth, hypertension during pregnancy, polyhydramnios, preterm birth < 37 weeks, congenital anomalies, gestational loss in the second trimester, and induction delivery), delivery complications (caesarean section, obstetric trauma, intrapartum death, postpartum complications, and maternal admission), and neonatal complications (Apgar test < 3 at 1 min, Apgar test < 3 at 5 min, arterial pH < 7 at birth, venous pH < 7 at birth, neonatal death, neonatal intensive care unit (NICU) admission, neonatal hypoglycemia, neonatal hypocalcemia, neonatal hyperbilirubinemia, neonatal polycythemia, and perinatal death).

The sample size was estimated taking into account the impact of screening tests on GDM and large-for-gestational-age prevalences. The GDM prevalence in our population is estimated at 12% with an error of up to 4% [23]. With a 95% confidence level, the sample size would be 1015 participants per group (2030 cases in total). The large-for-gestational-age fetal prevalence in our population is estimated at 16.5% with an error of up to 4% [24]. With a 95% confidence level, the sample size would be 1324 participants per group (2648 cases in total). Therefore, the required target sample size is 3000 pregnant individuals, which allows for up to 10% of cases with incomplete or missing data.

Descriptive data are expressed as the median and interquartile range (IQR), or as absolute and relative frequencies, as appropriate. Variables were compared using the *t*-test or the Fisher’s exact test. A multivariable logistic regression model was fitted for each composite variable and for each outcome variable, using the composite or outcome variables as the dependent variable and the screening group (standard screening vs. alternative screening) as the independent variable, adjusted for obesity, maternal age, metabolic control, ethnicity, smoking status, family history of diabetes, previous GDM, chronic hypertension, parity, and site of inclusion.

The analysis was conducted for the entire sample and only in the group diagnosed with GDM. The diagnostic rate for each trimester with each method was compared using a proportionality test [25].

The odds ratio, 95% confidence interval, and *p* values are reported. All analyses were conducted using the software R, version 4.3 [and SPSS for Windows 19.0 statistical package].

## 3. Results

A total of 4811 pregnant individuals were considered for this study. Of those, 1071 were excluded: 274 due to pregestational diabetes, 182 due to multiple gestations, and 615 due to the unavailability of perinatal outcomes. Therefore, a total of 3740 pregnant individuals were included in this study: data were collected from 1543 pregnant individuals in the standard screening group and from 2197 in the alternative screening group.

### 3.1. Whole Population Analysis

In the whole population analysis, some differences in demographics were found (Table 1). Although maternal age, parity, and the number of preterm births were similar in both groups (*p* = 0.220, *p* = 0.199 and *p* = 0.131, respectively), there were statistically significant differences in the body mass index (BMI), with pregnant individuals having a slightly higher BMI in the alternative screening group (*p* = 0.002). We also found that individuals in the standard screening group were more frequently nulliparous (*p* = 0.001), had a higher prevalence of tobacco use during pregnancy (*p* < 0.0001), and chronic HTA prior to pregnancy (*p* = 0.024). In addition, there were some statistically significant differences in family history of DM and prior GDM between both groups (*p* = 0.007 and *p* < 0.0001, respectively), probably due to the high number of missing data in the standard screening group. There were also statistically significant differences for ethnicity between both groups. A higher rate of GDM risk factors in the first trimester was observed in the alternative screening group (*p* = 0.024). However, the standard screening group had a higher GDM diagnostic rate (10.8% vs. 6.9%, *p* < 0.0001).

The primary outcome (LGA rate) was similar between groups: 200/1543 (13.0%) vs. 303/2197 (13.8%) (OR: 1.74; 95% CI: 0.74–4.10; *p* = 0.672).

An adjusted analysis considering maternal obesity (BMI ≥ 30 kg/m^2^), parity, ethnicity, smoking status, family history of DM, previous GDM, GMD risk factors in the first trimester, previous HTA, the GDM screening method, and the site of inclusion was performed (Table 2A,B). There were no differences between groups for the composite adverse outcomes of pregnancy complications (OR: 1.11; 95% CI: 0.91–1.36; *p* = 0.312), delivery complications (OR: 0.95 (95% CI: 0.75–1.19; *p* = 0.642), and neonatal complications (OR: 1.28; 95% CI: 0.94–1.75; *p* = 0.118).

### 3.2. GDM Group Analysis

The demographic characteristics of individuals diagnosed with GDM were analyzed and compared (Table 3). There were no differences between groups for maternal age, maternal weight increase during pregnancy, previous preterm births, tobacco use and drug abuse during pregnancy, previous GDM, GDM risk factors in the first trimester, chronic HTA prior to pregnancy, and metformin treatment. However, there were some statistically significant differences in the family history of DM between groups and there was a higher rate of nulliparity in the standard screening group (*p* = 0.003). There were also statistically significant differences between groups for metabolic control, with a poorer metabolic control (*p* < 0.001), higher need for basal insulin (*p* = 0.015), and multiple insulin doses (*p* = 0.002) in the alternative screening group.

In individuals diagnosed with GDM, the primary outcome (LGA rate at birth) was similar between groups: 13/166 (7.8%) vs. 15/151 (9.9%) (*p* = 0.534) (OR: 1.24; 95% CI: 0.51–3.09; *p* = 0.634).

An analysis adjusting for maternal obesity (BMI ≥ 30), parity, ethnicity, smoking status, family history of GDM, previous GDM, GMD risk factors in the first trimester, previous HTA, the GDM screening method, and the site of inclusion was performed for composite outcomes (Table 4A,B). When comparing the standard and alternative screening groups, there were no differences in the composite adverse outcomes for pregnancy complications (OR: 1.57; 95% CI: 0.84–2.98; *p* = 0.159), delivery complications (OR: 1.21; 95% CI: 0.63–2.35; *p* = 0.565), and neonatal complications (OR: 1.35; 95% CI: 0.61–3.04; *p* = 0.455).

In order to investigate the impact of metabolic control in pregnancy and neonatal outcomes in individuals diagnosed with GDM and show their impact on previous results, we studied adverse outcomes in terms of pregnancy complications, perinatal complications, and neonatal complications, adjusting for metabolic control and insulin treatment variables (number of individuals needing insulin and dose) (Table 5). After adjusting for confounders, no differences were found between the two groups (BMI, parity, ethnicity, smoking status, family history of DM, previous GDM, risk factors for GDM in the first trimester, previous HTA, GDM screening method, and site of inclusion) in composite adverse outcomes for pregnancy complications (OR: 0.51; 95% CI: 0.22–1.14; *p* = 0.109), delivery complications (OR: 0.98; 95% CI: 0.49–1.99; *p* = 0.955), and neonatal complications (OR: 0.83; 95% CI: 0.35–1.96; *p* = 0.664).

In order to study the financial impact of the alternative screening method, we analyzed the procedure and costs, including expenses for consumables, equipment, and personnel, of each screening method (Table 6). There were statistically significant differences between the costs of the two screening methods, with the alternative screening method being cheaper. The mean cost of the alternative screening method was EUR 46.0 (SD: 22.3 SD), as compared to EUR 85.6 (SD: 67.5) for the standard screening method.

## 4. Discussion

### 4.1. Findings

The DIABECOVID study assessed the impact of using an alternative method for GDM screening during the COVID-19 pandemic as compared to the standard method. The research involved two cohorts of pregnant individuals from different healthcare sites across Catalonia, Spain, included between April 2020 and April 2022. One cohort underwent the standard screening method for GDM, while the other cohort underwent the alternative screening method based on plasma glucose and HbA1c levels (fasting or non-fasting). The aim was to assess the impact of using alternative diagnostic criteria for gestational diabetes mellitus (GDM) during the COVID-19 pandemic on GDM prevalence and obstetrical and perinatal outcomes as compared with the standard screening method, as well as the financial impact.

The findings revealed several important insights. Firstly, the standard screening group had a higher GDM prevalence than the alternative screening group (10.8% vs. 6.9%). Interestingly, the primary outcome (LGA rate) was similar between the two groups. An adjusted analysis showed no significant differences in the composite adverse outcomes for pregnancy, delivery, and neonatal complications between both groups.

Again, the sub-analysis of individuals diagnosed with GDM showed no significant differences in the LGA rates or composite adverse outcomes for pregnancy, delivery, and neonatal complications between both groups. Although there was a higher need for insulin therapy in the alternative screening group, this did not translate into adverse outcomes. This suggests that the alternative screening method, being a less specific and less sensitive method, is not able to identify patients with less severe GDM.

Furthermore, a cost analysis revealed that the alternative screening method was cheaper than the standard screening method. The average cost per patient for the alternative screening was EUR 46.0, while the cost for the standard screening was EUR 85.6.

This study suggests that the alternative GDM screening criteria, implemented during the COVID-19 pandemic, has a lower GDM diagnostic rate than the standard method. However, the obstetrical, delivery, and neonatal outcomes for both groups remained comparable. The alternative screening method had also a lower diagnostic sensitivity. This research highlights the adaptability of diagnostic screening approaches in response to unique circumstances, such as those of a pandemic, and the importance of assessing the impact of implementing a new screening method on both healthcare outcomes and costs.

### 4.2. Impact and Correlation with the Literature

Elevated fasting glucose levels in the first trimester are consistently linked to a higher risk of large-for-gestational-age babies, macrosomia, and cesarean section. This association persists even after excluding individuals with a basal fasting glucose higher than 105 mg/dL [26]. The rise in fasting plasma glucose during early pregnancy [27], driven by placental hormones inducing insulin resistance, challenges the appropriateness of performing GDM screening (normally performed at 24–28 weeks) before 20 weeks of gestation.

Moreover, some studies suggest that overtreatment may lead to small-for-gestational-age infants and adverse fetal metabolic programming, contributing to cardiovascular diseases later in life for both the mother and the infant [28,29]. Thus, a key challenge is establishing evidence-based glycemic cut-off values in the first trimester for early GDM diagnosis.

Few studies have explored the addition of HbA1c to the diagnostic criteria for patients with a high GDM risk. Amylidi et al. found that first-trimester HbA1c levels were predictive of GDM, similar to the standard screening method. Adding HbA1c to the GDM diagnostic criteria with a cut-off value of 5.4% in the first trimester may have identified 18% of GDM cases in the second trimester, and a cut-off value of 5.7% may have identified 32% of cases [30]. These authors suggest that a HbA1c > 6% predicts GDM with 100% accuracy and warrants treatment. In the Roeder intervention study [31], the group with fasting glucose ≥ 5.1 mmol/l (92 mg/dL) or HbA1c ≥ 5.7% had more individuals with high HbA1c than with high fasting glucose. However, using HbA1c alone as the only criterion for GDM diagnosis lacks sufficient sensitivity or specificity, although a higher HbA1c in the first trimester is significantly associated with an increased risk of GDM [32]. In the context of the COVID-19 pandemic, González González et al. showed that combining HbA1c with fasting glucose during early pregnancy maintained comparable diagnostic rates to the standard method while minimizing hospital visits [33]. These findings underscore the need for further studies to refine cut-offs and optimize the use of HbA1c in GDM screening, as supported by our study’s results.

In a prospective study conducted in a Spanish population, adding HbA1c to the diagnostic criteria in the first-trimester and second-trimester GDM screenings without additional criteria, a cut-off value of <4.7% for HbA1c had a 95.3% negative predictive value for GDM in the first trimester, potentially reducing screening costs by 5.7% [34]. However, this cut-off value had only a 14% positive predictive value in the second trimester. With a higher cut-off value for HbA1c, such as 5.6%, the positive predictive value in the second trimester increased to 31.6%, with a maintained negative predictive value of 89.8%. The results of that study highlight the importance of optimizing first-trimester screening criteria for early GDM prediction, including glycemic criteria, to avoid unnecessary tests and costs.

Several studies show that glucose levels at different points of the OGTT test in the first trimester are significantly higher in subjects who later develop GDM in the second trimester [35,36]. This suggests that adding fasting glucose levels to the diagnostic criteria may also have a high negative predictive value, potentially reducing the number of subjects requiring GDM screening. There is no consensus on a particular fasting glucose threshold for early GDM diagnosis. Therefore, well-designed trials are needed to establish a fasting glucose cut-off value and determine the risk of adverse maternal and fetal outcomes [37].

Basal fasting glucose decreases from week 5 to week 19 of gestation in all pre-gestational BMI groups [38]. However, being overweight or obese before pregnancy is associated with increased fasting glucose between 19 and 24 gestational weeks. Therefore, combining glycemic criteria with clinical criteria, such as a maternal pre-gestational BMI, may also help to develop a reliable diagnostic algorithm.

During the COVID-19 pandemic, Spanish healthcare sites approved a consensus document allowing GDM diagnosis using only glucose and HbA1c [11]. Before the pandemic, a Spanish study evaluating diagnostic criteria and neonatal outcomes in 3200 subjects using only glucose (1t-FPG ≥ 100 mg/dL) for GDM or gestational hyperglycemia diagnosis in the first trimester found that 79.5% of pregnant individuals with early GDM would not have been diagnosed using the OGTT and two-step screening method [33]. On the other hand, a Spanish cohort showed that GDM prevalence during the COVID-19 pandemic, when using the alternative diagnostic criteria (glucose and HbA1c measurements), was similar to GDM prevalence in 2019, when the standard diagnostic criteria was used. Although individuals referred for GDM screening during the pandemic had more GDM risk factors, obstetrical and perinatal outcomes were comparable to those before the pandemic [15].

Thus, our study provides more evidence suggesting that including glucose and HbA1c in the same blood test improves the predictive value for GDM.

Subsequent research is essential to determine optimal cut-off values, formulate algorithms that include glucose and HbA1c, and consider additional factors, such as ethnicity or BMI. This comprehensive approach aims to further refine the predictive algorithm.

### 4.3. Strengths

The study on the impact of GDM diagnostic criteria during the COVID-19 pandemic (DIABECOVID STUDY) has several strong points which are of relevance during a pandemic: the study addresses a critical issue during a pandemic; compares two methods (standard method and alternative method adopted during the pandemic) for GDM screening through a comprehensive analysis; and provides comprehensive outcome measures and adjusts for confounders, such as maternal age, BMI, parity, ethnicity, smoking status, and family history. This enhances the reliability of the findings. In addition to assessing clinical outcomes, the study also examines the financial impact of the alternative screening method through a cost analysis. The data show that the alternative screening method is cheaper, providing valuable information for healthcare providers and policymakers.

### 4.4. Limitations

Despite its strengths, the study on the impact of an alternative GDM screening method during the COVID-19 pandemic (DIABECOVID STUDY) has some limitations. On one hand, the study involved retrospective data collection during the initial phase of the COVID-19 pandemic. This may have introduced recall bias and limit the availability of complete data, potentially affecting the accuracy of results. There was also a selection bias; the decision of whether to use the standard or alternative diagnostic criteria was made by each healthcare site. This may have introduced selection bias, as sites that adopted the alternative method might have had different patient populations or policies that may have influenced outcomes. On the other hand, the study had incomplete and missing data: the exclusion of some participants due to missing perinatal outcomes may have affected the sample’s representativeness, potentially introducing bias if certain individuals were more likely to be excluded. Regarding missing data, it must be noted that some relevant data, such as a family history of diabetes, had a high number of missing data, which may have impacted the accuracy of the results. Lastly, the study has limited applicability, as it was conducted in only one Spanish region (Catalonia), and the results may not be applicable to other regions with different patient populations, healthcare practices, and pandemic situations.

Another limitation lies in the financial impact analysis, where only the screening costs were considered. The additional costs associated with overdiagnosis in the standard screening method, including the number of medical visits, indirect or quality-of-life costs for pregnant individuals, and pregnancy outcomes, were not taken into account.

### 4.5. Further Research

The COVID-19 pandemic revealed the challenges pregnant individuals face in accessing GDM screening. Alternative screening methods, such as those implemented during the pandemic, provide an opportunity to rethink and optimize the GDM screening process.

One recent randomized controlled trial [39], comparing lower (92 mg/dL) versus higher (99 mg/dL) glycemic criteria for GDM diagnosis, showed that using the lower glycemic criteria led to a higher diagnostic rate than using the higher glycemic criteria; therefore, the use of healthcare services was greater in the group with lower glycemic criteria. Overall, the risks of giving birth to a large-for-gestational-age infant and other infant or maternal complications were not lower for the group with lower glycemic criteria. Although using the lower glycemic criteria led to a greater proportion of individuals receiving a GDM diagnosis and treatment, it did not lead to clear health benefits. However, using the lower glycemic criteria led to a greater use of healthcare services. The authors found more infants with hypoglycemia warranting treatment in the group with lower glycemic criteria. Neonatal hypoglycemia is associated with adverse neurodevelopmental outcomes later in life, so a follow-up will be needed to determine whether GDM diagnosis and treatment leads to benefits or adverse outcomes later in life. Also, cost analyses will be necessary to aid decision-making.

New prospective clinical trials should aim to compare different GDM screening methods, including universal screening, risk factor-based screening, and alternative screening, while considering their impact on maternal and neonatal outcomes, healthcare costs, and healthcare system adaptability during crises. There is also a need to collect patient-reported outcomes and quality-of-life questionnaires to determine whether the new screening method has any positive impact on patients. It is likewise important to consider new studies to evaluate oral agents as a therapeutic option in women with GDM [40].

## 5. Conclusions

The DIABECOVID study found that the alternative screening method used during the pandemic resulted in a lower GDM prevalence, although obstetric and perinatal outcomes were comparable between the two groups. Notably, glycemic control and insulin treatment did not significantly differ between individuals receiving a GDM diagnosis and those who did not. However, the alternative screening method had a lower GDM diagnostic rate. Despite this, our study suggests that the alternative screening method for GDM screening can be effective in a population with a 12% incidence of GDM, as well as cheaper. These results need to be confirmed in prospective randomized controlled trials.

## Figures and Tables

**Table 1 diagnostics-15-00189-t001:** Demographic and general characteristics of the whole population of the study, classified according to screening method.

	Standard Screening Method *n* = 1543	Alternative Screening Method *n* = 2197	*p* Value
Maternal age (years)	32.2 (5.7)	32.2 (5.9)	0.220
Obesity [BMI (kg/m^2^) ≥ 30] No Yes No data	1260 (81.7%) 237 (15.3%) 46 (3.0%)	1729 (78.7%) 431 (19.6%) 37 (1.7%)	0.002
Previous term birth No One Two Three Four Five or more	748 (48.5%) 514 (33.3%) 189 (12.2%) 68 (4.4%) 16 (1.0%) 8 (0.5%)	997 (45.4%) 732 (33.3%) 314 (14.3%) 104 (4.7%) 32 (1.5%) 18 (0.8%)	0.199
Previous preterm birth No One Two Three or more	1476 (95.7%) 63 (4.1%) 4 (0.3%) 0 (0%)	2065 (93.9%) 121 (5.5%) 10 (0.5%) 1 (0.1%)	0.131
Parity Nulliparous Multiparous	674 (43.7%) 868 (56.3%)	845 (38.5%) 1352 (61.5%)	0.001
Ethnicity Caucasian African American Asiatic Maghreb Hindu Latin American Others No data	873 (56.8%) 106 (6.9%) 35 (2.3%) 237 (15.4%) 35 (2.3%) 241 (15.7%) 10 (0.7%) 6 (0.4%)	1298 (59.1%) 59 (2.7%) 33 (1.5%) 231 (10.5%) 79 (3.6%) 422 (19.2%) 67 (3.0%) 8 (0.4%)	<0.0001
Tobacco use during pregnancy No Yes	1310 (84.9%) 233 (15.1%)	1952 (88.9%) 245 (11.1%)	<0.0001
Drug abuse during pregnancy No Yes No data	1512 (98.0%) 30 (1.9%) 1 (0.1%)	2157 (98.2%) 37 (1.7%) 3 (0.1%)	0.557
Family history of DM No Yes No data	803 (52.0%) 66 (4.3%) 674 (43.7%)	1725 (78.5%) 177 (8.1%) 295 (13.4%)	<0.0001
Previous GDM No Yes No data	1412 (91.5%) 46 (3.0%) 85 (5.5%)	2048 (93.2%) 74 (3.4%) 75 (3.4%)	0.007
Risk factors for GMD in first trimester No Yes	1267 (82.1%) 276 (17.9%)	1689 (76.9%) 508 (23.1%)	<0.0001
Chronic HTA before pregnancy	24 (1.6%)	17 (0.8%)	0.024
GDM diagnosis rate	166 (10.8%)	152 (6.9%)	<0.0001

BMI: body mass index, GDM: gestational diabetes mellitus, HTA: hypertension arterial. Data are shown as the median and standard deviation for quantitative variables; as number of cases and percentage for qualitative variables. The *p* value was considered statistically significant at <0.05. Tests used were the Chi-square test for qualitative variables and the Student’s *t* test for quantitative variables.

**Table 2 diagnostics-15-00189-t002:** (**A**) Pregnancy complications, perinatal events, and neonatal morbidities in the whole population of the study, classified according to screening method. (**B**) Composite adverse outcomes including pregnancy complications, delivery complications, and neonatal complications classified according to screening method (reference category: standard screening method).

(A)				
Characteristic	Standard Screening Method *n* = 1543	Alternative Screening Method *n* = 2197	Unadjusted OR (95% CI)	Adjusted OR (95% CI)
Macrosomia at birth	96 (6.2%)	122 (5.6%)	0.89 (0.67–1.17)	0.98 (0.65–1.48)
LGA at birth (>P90)	200 (13.0%)	303 (13.8%)	1.07 (0.89–1.30)	0.92 (0.68–1.23)
Hypertension during pregnancy	17 (1.1%)	25 (1.1%)	1.3 (0.98–1.74)	1.03 (0.47–2.26)
Polyhydramnios	16 (1.0%)	16 (0.7%)	0.71 (0.35–1.43)	0.86 (0.27–2.78)
Preterm birth (PB) < 37 weeks	90 (5.8%)	201 (9.1%)	1.73 (1.34–2.25)	1.16 (0.78–1.73)
Spontaneous PB < 37 weeks	42 (2.7%)	98 (4.5%)	2.11 (1.46–3.09)	1.64 (0.96–2.82)
Gestational loss in 2nd trimester	9 (0.6%)	17 (0.8%)	1.35 (0.61–3.17)	0.49 (0.16–1.47)
Induction delivery rate	524 (34.0%)	824 (37.5%)	1.17 (1.02–1.34)	0.91 (0.74–1.12)
Cesarean section rate	342 (22.3%)	520 (23.7%)	1.17 (1.00–1.36)	0.81 (0.63–1.03)
Obstetric trauma	15 (1%)	19 (0.9%)	0.90 (0.46–1.80)	1.79 (0.81–3.95)
Intrapartum fetal death	2 (0.1%)	2 (0.1%)	0.71 (0.09–5.93)	0.82 (0.05–14.08)
Postpartum complications	5 (0.3%)	7 (0.3%)	0.98 (0.31–3.33)	0.75 (0.55–1.07)
Maternal admission to hospital	125 (8.1%)	188 (9.1%)	1.13 (0.90–1.44)	1.07 (0.82–1.40)
Apgar test < 3 at 1 min	20 (1.3%)	26 (1.2%)	0.97 (0.54–1.76)	0.86 (0.35–2.19)
Apgar test < 3 at 5 min	13 (0.8%)	12 (0.5%)	0.69 (0.31–1.52)	0.34 (0.12–1.10)
Neonatal death	4 (0.3%)	6 (0.3%)	1.07 (0.3–4.18)	0.63 (0.09–4.62)
NICU admission	63 (4.1%)	114 (5.2%)	1.33 (0.98–1.84)	0.78 (0.47–1.31)
Neonatal hypoglycemia	41 (2.7%)	57 (2.6%)	1.01 (0.67–1.53)	1.28 (0.73–2.26)
Perinatal death	50 (3.2%)	79 (3.6%)	1.15 (0.81–1.66)	0.95 (0.55–1.65)
(**B**)				
**Characteristic**	**Unadjusted Model**		**Adjusted Model ***	
Composite of pregnancy complications ^1^	OR: 1.38 (95% CI: 1.17–1.62) *p* < 0.001		OR: 1.11 (95% CI: 0.91–1.36) *p* = 0.312	
Composite of delivery complications ^2^	OR: 1.15 (95% CI: 0.95–1.39) *p* = 0.159		OR: 0.95 (95% CI: 0.75–1.19) *p* = 0.642	
Composite of neonatal complications ^3^	1.05 (0.83–1.33) *p* = 0.718		OR: 1.28 (95% CI: 0.94–1.75) *p* = 0.118	

^1^ Includes macrosomia at birth, large for gestational (LGA) at birth (P ≥ 90), hypertension during pregnancy, polyhydramnios, preterm birth < 37 weeks, gestational loss during second trimester and induction delivery rate. ^2^ Includes caesarean section rate, obstetric trauma, intrapartum death, postpartum complications, and maternal admission. ^3^ Includes Apgar test < 3 at 1 min, Apgar test < 3 at 5 min, neonatal death, NICU admission, neonatal hypoglycemia, and perinatal death. * Logistic regression models adjusted for maternal obesity (BMI ≥ 30), parity, ethnicity, smoking status, previous GDM, risk factors for GMD in first trimester, previous HTA, and site of inclusion.

**Table 3 diagnostics-15-00189-t003:** Demographic and general characteristics of individuals with diagnosis of gestational diabetes according to screening method.

	Standard Screening Method *n* = 166	Alternative Screening Method *n* = 151	*p* Value
Maternal age (years)	33.9 (5.2)	34.0 (5.5)	0.929
Obesity [BMI (kg/m^2^) ≥ 30] No Yes No data	111 (66.9%) 54 (32.5%) 1 (0.6%)	74 (49.0%) 74 (49.0%) 3 (2.0%)	0.002
Maternal weight increase during pregnancy (kg)	6.4 (0.5)	6.6 (0.5)	0.131
Previous term birth No One Two Three Four Five or more	74 (44.6%) 45 (27.1%) 30 (18.1%) 12 (7.2%) 3 (1.8%) 2 (1.2%)	39 (25.8%) 55 (36.4%) 35 (23.2%) 18 (11.9%) 1 (0.7%) 3 (2.0%)	0.016
Previous preterm birth No One Two Three or more	157 (94.6%) 9 (5.4%) 0 (0%) 0 (0%)	141 (93.4%) 8 (5.3%) 2 (1.3%) 0 (0%)	0.331
Parity Multiparous Nulliparous	64 (38.6%) 102 (61.4%)	35 (23.2%) 116 (76.8%)	0.003
Ethnicity Caucasian African American Asiatic Maghreb Hindu Latin American Others	84 (50.6%) 11 (6.6%) 4 (2.4%) 38 (22.9%) 6 (3.6%) 21 (12.7%) 2 (1.2%)	58 (38.4%) 7 (4.6%) 1 (0.7%) 21 (13.9%) 17 (11.3%) 421(27.2%) 6 (4.0%)	<0.0001
Tobacco use during pregnancy No Yes	147 (88.6%) 19 (11.4%)	135 (89.4%) 16 (10.6%)	0.809
Drug abuse during pregnancy No Yes	165 (99.4%) 1 (0.6%)	150 (99.3%) 1 (0.7%)	0.946
Family history of DM No Yes No data	74 (44.6%) 25 (15.1%) 67 (40.4%)	97 (64.2%) 37 (24.5%) 17 (11.3%)	<0.0001
Previous GDM No Yes No data	131 (78.9%) 31 (18.7%) 4 (2.4%)	116 (76.8%) 29 (19.2%) 6 (4.0%)	0.716
Risk factors of GDM in first trimester No Yes	95 (57.2%) 71 (42.8%)	84 (55.6%) 67 (44.4%)	0.774
Chronic HTA before pregnancy No Yes	152 (91.6%) 14 (8.4%)	132 (87.4%) 19 (12.6%)	0.227
Glycemic control No control Inadequate control Suboptimal control Optimal control	5 (3.1%) 10 (6.3%) 35 (21.9%) 110 (68.8%)	16 (11%) 32 (21.9%) 30 (20.5%) 68 (46.6%)	<0.0001
Basal insulin No Yes	107 (64.5%) 59 (35.5%)	77 (51%) 74 (49%)	0.015
Insulin treatment with multiple doses No Yes	118 (71.1%) 48 (28.9%)	82 (54.3%) 69 (45.7%)	0.002
Metformin treatment No Yes	164 (98.8%) 2 (1.2%)	146 (96.7%) 5 (3.3%)	0.202

Data are shown as the median and standard deviation for quantitative variables; as the number of cases and percentage for qualitative variables. *p* value was considered statistically significant at <0.05. Tests used were the Chi-square test for qualitative variables and the Student’s *t* test for quantitative variables.

**Table 4 diagnostics-15-00189-t004:** (**A**) Pregnancy complications, perinatal events, and neonatal morbidities in individuals diagnosed with gestational diabetes, classified according to screening method. (**B**) Composite adverse outcomes considering pregnancy complications, delivery complications, and neonatal complications in individuals diagnosed with gestational diabetes according to screening method (reference category: standard screening method).

(A)				
Characteristic	Standard Screening Method *n* = 166	Alternative Screening Method *n* = 152	Unadjusted OR (95% CI)	Adjusted OR (95% CI)
Macrosomia at birth	13 (7.8%)	15 (9.9%)	1.29 (0.59–2.84)	1.19 (0.35–4.09)
LGA at birth (>P90)	36 (21.7%)	48 (31.8%)	1.67 (1.01–2.77)	1.74 (0.74–4.10)
Hypertension during pregnancy	10 (6.0%)	17 (11.2%)	1.96 (0.88–4.59)	1.43 (0.40–5.09)
Polyhydramnios	7 (4.2%)	4 (2.6%)	0.61 (0.16–2.08)	0.50 (0.07–3.30)
Preterm birth PB < 37 weeks	11 (6.7%)	15 (10.4%)	1.63 (0.73–3.76)	1.17 (0.41–3.38)
Gestational loss on second trimester	1 (0.6%)	0 (0%)	-	-
Induction delivery rate	78 (47.0%)	94 (61.8%)	1.83 (1.17–2.87)	1.57 (0.88–2.82)
Caesarean section rate	42 (25.5%)	44 (30.1%)	1.26 (0.77–2.08)	0.72 (0.37–1.4)
Obstetric trauma	4 (2.4%)	4 (2.6%)	1.09 (0.25–4.7)	1.16 (0.24–5.60)
Intrapartum fetal death	0 (0%)	1 (0.7%)	-	-
Postpartum complications Maternal sepsis	12 (7.2%)	20 (13.2%)	0.51 (0.24–1.08)	0.38 (0.15–0.91)
Maternal Admission	15 (9.1%)	19 (13.3%)	1.53 (0.75–3.18)	1.71 (0.71–4.19)
Apgar test < 3 at 1 min	3 (1.8%)	1 (0.7%)	0.38 (0.02–2.99)	-
Apgar test < 3 at 5 min	2 (1.2%)	1 (0.7%)	0.57 (0.03–6.01)	-
Neonatal death	1 (0.6%)	2 (1.3%)	2.2 (0.21–47.62)	-
NICU admission	6 (3.6%)	8 (5.3%)	1.52 (0.52–4.73)	1.25 (0.27–5.76)
Neonatal hypoglycemia	12 (7.2%)	15 (9.9%)	1.45 (0.66–3.26)	1.27 (0.48–3.37)
Perinatal death	8 (4.8%)	5 (3.3%)	0.69 (0.2–2.1)	0.24 (0.4–1.34)
(**B**)				
**Characteristic**	**Unadjusted Model**		**Adjusted Model ***	
Composite of pregnancy complications ^1^	OR: 1.41 (95% CI: 0.82–2.43) *p* = 0.216		OR: 1.57 (95% CI: 0.84–2.98) *p* = 0.159	
Composite of delivery complications ^2^	OR: 1.14 (95% CI: 0.65–2.01) *p* = 0.640		OR: 1.21 (95% CI: 0.63–2.35) *p* = 0.565	
Composite of neonatal complications ^3^	OR: 1.14 (95% CI: 0.65–2.01) *p* = 0.640		OR: 1.35 (95% CI: 0.61–3.04) *p* = 0.455	

^1^ Includes macrosomia at birth, large for gestational (LGA) at birth (P ≥ 90), hypertension during pregnancy, polyhydramnios, preterm birth < 37 weeks, gestational loss 2T, and induction delivery rate. ^2^ Includes caesarean section rate, obstetric trauma, intrapartum death, postpartum complications, and maternal admission. ^3^ Includes Apgar test < 3 at 1 min, Apgar test < 3 at 5 min, neonatal death, NICU admission, neonatal hypoglycemia, and perinatal death. * Logistic regression models adjusted for maternal obesity (IMC ≥ 30), parity, ethnicity, smoking status, previous GDM, risk factors for GMD in 1T, previous HTA, and center of inclusion.

**Table 5 diagnostics-15-00189-t005:** Composite adverse outcomes considering pregnancy complications, delivery complications, and neonatal complications correlated with metabolic control (and insulin variables) in individuals diagnosed with gestational diabetes.

Characteristic	Adjusted Model (95% CI) *
Composite for pregnancy complications ^1^	0.51 (0.22–1.14)
Composite for delivery complications ^2^	0.98 (0.49–1.99)
Composite for neonatal complications ^3^	0.83 (0.35–1.96)

^1^ Includes macrosomia at birth, large for gestational (LGA) at birth (P ≥ 90), hypertension during pregnancy, polyhydramnios, preterm birth < 37 weeks, gestational loss 2T, and induction delivery rate. ^2^ Includes caesarean section rate, obstetric trauma, intrapartum death, postpartum complications, and maternal admission. ^3^ Includes Apgar test < 3 at 1 min, Apgar test < 3 at 5 min, neonatal death, NICU admission, neonatal hypoglycemia, and perinatal death. * Logistic regression models adjusted for maternal obesity (BMI ≥ 30), parity, ethnicity, smoking status, previous GDM, risk factors for GMD in the first trimester, previous HTA, screening method for gestational diabetes and site of inclusion.

**Table 6 diagnostics-15-00189-t006:** Cost of each method (euros per individual).

	Standard Screening Method	Alternative Screening Method	*p* Value
Total Cost	*n* = 1543	*n* = 2197	
Mean (SD)	85.6 (67.5)	46.0 (22.3)	<0.001

Three aspects were considered for the cost: consumables (catheter, gases, vacutainer, tubes), laboratory (equipment), and personnel (nurses or midwives performing the tests).

## Data Availability

The original contributions presented in this study are included in the article. Further inquiries can be directed to the corresponding author.

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
