# Peer review of "Evaluation of an Alternative Screening Method for Gestational Diabetes Diagnosis During the COVID-19 Pandemic (DIABECOVID STUDY): An Observational Cohort Study"

_diagnostics, 2025, doi:10.3390/diagnostics15020189_

Round 1
Reviewer 1 Report
Comments and Suggestions for Authors
Abstract
GMD in line 42- full form? Is it a typing error throughout in the manuscript? Please rectify.
Among individuals diagnosed 56 with GDM, the large for gestational age rate was similar between groups: 13/166 (7.8 %) vs 15/151 57 (9.9 %). (Line 56 and 57): what is pvalue?
GDM prevalence was lower in the alternative screening group than in the standard 64 screening group (line 64): do you mean to say that the standard method is overdiagnosing GDM prevalence?
Introduction:
· Line 77- what kind of complexities?
Methods
· When the COVID-19 pandemic started, an alternative Spanish method for GDM screening was developed and published in May 115 2020. (Line 114-115): Provide reference
· Patient consent was waived 118 for this cohort due to its retrospective nature. (Line 118): Was ethics obtained? Provide registration number and date of approval.
A cohort of pregnant individuals undergoing an alternative GMD screening (based 123 on plasma glucose and HbA1c, fasting or non-fasting) due to the COVID-19 pandemic was compared with a cohort of pregnant individuals undergoing the standard GMD screening. (Line 123-125): N=?
decided to either maintain the standard GDM screening protocol or switch to an alternative method 131 proposed by different medical societies.(Lines 13-131)Would this not create a bias if all centers use different standard protocols?
Please re-write the methods section to past tense. E.g. In the first trimester, screening is performed following the Catalan regional guidelines (Line 136): are not is.
Lines 135-148 need appropriate and sufficient referencing.
Third-trimester screening is performed only in cases with large for gestational age (LGA) fetus (Line 146): how did you diagnose LGA fetuses?
preferably non-fasting due to feasibility (Line 154): what kind of feasibility?
The group diagnosed with GDM 163 underwent a specific diet and glucose self-monitoring (Line 163): N=?
RedCap (Line 168): describe in brief . also provide validity and reliability of the tool used
Lines 171-194 need appropriate and sufficient referencing.
Lines 208-215: provide reference for prevalence of LGA and GDM
Line 221: How did you account for metabolic control?, why did you account for chronic hyper-tension, there was no mention of this before in the manuscript.
Results
HTA (Line 243): fullform?
probably due to the high number of missing data in the standard screening group (Line 246): how did you account for this error?
Table 2 A, 4 A and B, 5: mention p values
None of the neonatal complications have been explicitly mentioned. It will be worthwhile to describe them instead of just composite measures.
Discussion
The alternative screening method had also a lower diagnostic sensitivity (Line 379): how was this assessed?
The discussion limits comparison to other studies. HbA1c has not been shown in results but has been discussed in the discussion section. More research studies need to be added.
Author Response
We sincerely thank the reviewer for their valuable feedback and insightful comments on our manuscript. The constructive suggestions of the reviewer have contributed significantly to enhancing the clarity and rigour of our study. We sincerely appreciate recognising the reviewer’s efforts to address limitations and the relevance of the study's findings.
We have answered all the Reviewer 1 comments as follows:
Abstract
GMD in line 42- full form? Is it a typing error throughout in the manuscript? Please rectify.
Yes, it should be GDM.
Among individuals diagnosed 56 with GDM, the large for gestational age rate was similar between groups: 13/166 (7.8 %) vs 15/151 57 (9.9 %). (Line 56 and 57): what is pvalue?
The p-value es > 0.05, and unadjusted OR is 1.29 (0.59-2.84), and adjusted OR is 1.19 (0.35-4.09).
GDM prevalence was lower in the alternative screening group than in the standard 64 screening group (line 64): do you mean to say that the standard method is overdiagnosing GDM prevalence?
We don’t know if the standard method is overdiagnosing GDM or if the alternative method is underdiagnosing; we probably cannot answer this question with this study.
Introduction:
- Line 77- what kind of complexities?
It refers to how the COVID-19 pandemic exacerbated an already challenging problem.
Traditional methods, like the oral glucose tolerance test (OGTT), became more challenging due to concerns about prolonged hospital stays and the risk of virus exposure. This necessitated the development of alternative diagnostic methods, which added another "layer of complexity" to the existing debates and challenges around GDM screening protocols.
Methods
- When the COVID-19 pandemic started, an alternative Spanish method for GDM screening was developed and published in May 115 2020. (Line 114-115): Provide reference
The reference number 7 (introduction, line 85) has been added.
- Patient consent was waived 118 for this cohort due to its retrospective nature. (Line 118): Was ethics obtained? Provide registration number and date of approval.
The study was conducted in accordance with the Declaration of Helsinki, and was approved by the Vall d’Hebron Barcelona Hospital Campus Ethics Committee (VHEC) PR(AMI)381/2020 in October 2020. The Ethics Committee of each site also approved this study.
A cohort of pregnant individuals undergoing an alternative GMD screening (based 123 on plasma glucose and HbA1c, fasting or non-fasting) due to the COVID-19 pandemic was compared with a cohort of pregnant individuals undergoing the standard GMD screening. (Line 123-125): N=?
decided to either maintain the standard GDM screening protocol or switch to an alternative method 131 proposed by different medical societies.(Lines 13-131)Would this not create a bias if all centers use different standard protocols?
Unfortunately, the stratification of GDM screening methods introduced potential bias due to protocol variability across centres. However, this approach was the only feasible solution to ensure patient safety and maintain essential diagnostic processes during the COVID-19 pandemic.
To mitigate these issues, adjustments were made for variables such as obesity, parity, ethnicity, smoking status, family history of diabetes, prior GDM, first-trimester risk factors, and inclusion site. These efforts aimed to reduce the risk of bias associated with the variability in protocols across centres. However, it is acknowledged that these measures cannot eliminate the potential bias stemming from the non-standardized decision-making processes, as mentioned in the limitations section.
Please re-write the methods section to past tense. E.g. In the first trimester, screening is performed following the Catalan regional guidelines (Line 136): are not is.
Lines 135-148 need appropriate and sufficient referencing.
We have changed, as the reviewer suggested:
“The standard GDM screening consisted of a two-step format, and tests were conducted in the first, second, and third trimesters. 18 In the first trimester, screening was carried out following the Catalan regional guidelines, provided risk factors were detected. These risk factors included age >35 years, obesity (BMI ≥30 kg/m² or 27.5 kg/m² in Asian Americans), a first-degree family history of diabetes, and specific ethnic backgrounds (African Americans, Latino Americans, Pacific Islanders, Asian Americans). The test performed during the first trimester consisted of fasting glucose. If glucose levels exceeded 92 mg/dl, a 1-hour O’Sullivan test (OGTT) involving the ingestion of 50 g of glucose was administered. If glucose levels after one hour were ≥140 mg/dl, a 100-g OGTT was conducted. The diagnostic cut-off levels for fasting glucose in the standard GDM screening were ≥105 mg/dl for fasting, ≥190 mg/dl for 1-hour, ≥165 mg/dl for 2-hour, and ≥145 mg/dl for 3-hour glucose measurements (at least two of these thresholds were required for a diagnosis). Universal GDM screening was performed in the second trimester, between 24 and 28 gestational weeks. Third-trimester screening was limited to cases with large-for-gestational-age (LGA) fetuses, polyhydramnios, or when no screening had been conducted in the earlier trimesters.”
Third-trimester screening is performed only in cases with large for gestational age (LGA) fetus (Line 146): how did you diagnose LGA fetuses?
Large for gestational age (LGA) fetuses were diagnosed using customized growth curves, where LGA was defined as a birth weight above the 90th percentile (lines 176-178).
preferably non-fasting due to feasibility (Line 154): what kind of feasibility?
The feasibility here relates to reducing the burden on patients and healthcare systems by minimizing fasting tests' time and logistical challenges.
The group diagnosed with GDM 163 underwent a specific diet and glucose self-monitoring (Line 163): N=?
The group referred to includes all pregnant women diagnosed with GDM at the time of their diagnosis. All these women were initially placed on a specific diet and glucose self-monitoring regimen. Subsequently, a subset required insulin therapy based on their glucose control and clinical management guidelines.
RedCap (Line 168): describe in brief . also provide validity and reliability of the tool used
RedCap (Research Electronic Data Capture) is a secure, web-based application designed to support data capture for research studies. It provides an intuitive interface for validated data entry, audit trails for tracking data manipulation, and automated export procedures for seamless data downloads. RedCap is widely used in clinical and observational research due to its flexibility in managing various types of studies.
In terms of validity and reliability, RedCap is recognized for its robust data security, compliance with data protection standards (such as HIPAA for healthcare data), and its ability to maintain data integrity through validation rules, automated workflows, and role-based access. Studies evaluating RedCap have consistently demonstrated its reliability in data capture and its ability to reduce errors compared to traditional methods. These features make it a trusted tool for large-scale research projects across diverse domains.
Lines 171-194 need appropriate and sufficient referencing.
We have added the references as the reviewer suggested.
-American College of Obstetricians and Gynecologists. Gestational hypertension and preeclampsia: ACOG practice bulletin summary. Obstet Gynecol. 2020;135(6):1492–5.
-Barnsley Hospital NHS Foundation Trust. Polyhydramnios Clinical Guidelines [Internet]. 2023. Available from: https://www.barnsleyhospital.nhs.uk
-Papageorghiou AT, Kennedy SH, Salomon LJ, et al. International standards for early fetal size and pregnancy dating based on ultrasound measurement of crown-rump length in the first trimester. Ultrasound Obstet Gynecol. 2014;44(6):641–8.
Lines 208-215: provide reference for prevalence of LGA and GDM
We have added the references as the reviewer suggested.
-Duran, A. et al. Introduction of IADPSG criteria for the screening and diagnosis of gestational diabetes mellitus results in improved pregnancy outcomes at a lower cost in a large cohort of pregnant women: the St. Carlos Gestational Diabetes Study. Diabetes Care 37, 2442–50 (2014).
-Ricart, W. et al. Potential impact of American Diabetes Association (2000) criteria for diagnosis of gestational diabetes mellitus in Spain. Diabetologia 48, 1135–41 (2005).
Line 221: How did you account for metabolic control?,
Metabolic control was accounted for by analyzing glycemic control through categories such as "optimal control," "suboptimal control," and "inadequate control." Additionally, insulin treatment variables, including the number of individuals requiring insulin and the number of doses, were included in the analysis. This allowed for adjustments in outcomes related to pregnancy, delivery, and neonatal complications, ensuring that the impact of metabolic control was appropriately assessed and incorporated into the study results.
why did you account for chronic hyper-tension, there was no mention of this before in the manuscript. 2
Chronic hypertension, a well-established risk factor for adverse pregnancy outcomes, including gestational hypertension and preeclampsia, was considered in the analysis. Accounting for chronic hypertension was necessary to control for its potential confounding effects on outcomes such as gestational complications and neonatal health. While it may not have been explicitly detailed earlier in the manuscript, its inclusion in the statistical adjustments ensures a more accurate assessment of the impact of GDM screening methods on pregnancy outcomes. However, as the reviewer suggested, we can mention it in the methods.
Results
HTA (Line 243): fullform?
We have indicated that “HTA” should mean “hypertension”.
probably due to the high number of missing data in the standard screening group (Line 246): how did you account for this error?
The high number of missing data in the standard screening group was addressed by incorporating statistical adjustments in the analysis. Multivariable logistic regression models were used to adjust for critical confounders such as maternal age, BMI, parity, ethnicity, smoking status, and inclusion site.
Table 2 A, 4 A and B, 5: mention p values
P-values were > 0.05, so we chose not to include them to avoid adding information that might not be meaningful to the tables. However, we are happy to include them if the reviewer considers it necessary.
None of the neonatal complications have been explicitly mentioned. It will be worthwhile to describe them instead of just composite measures.
Discussion
The alternative screening method had also a lower diagnostic sensitivity (Line 379): how was this assessed?
The lower diagnostic sensitivity of the alternative screening method was assessed by comparing the GDM diagnostic rates between the standard and alternative methods. The study found that the alternative method identified fewer cases of GDM (6.9%) than the standard method (10.8%). This difference in diagnostic rates suggests that the alternative method had a lower sensitivity for detecting GDM, potentially missing cases of less severe glucose intolerance than the standard method could identify. These findings were supported by statistical analyses that adjusted for confounders.
The discussion limits comparison to other studies. HbA1c has not been shown in results but has been discussed in the discussion section. More research studies need to be added.
While HbA1c values may not have been explicitly highlighted in the results, their inclusion in the alternative screening method and their implications for diagnostic sensitivity and specificity were discussed in detail.
As the reviewer suggested, we added two sentences to the paragraph at the discussion, with additional studies focusing on the use of HbA1c for GDM diagnosis:
“In the context of the COVID-19 pandemic, González González et al. showed that combining HbA1c with fasting glucose during early pregnancy maintained comparable diagnostic rates to the standard method while minimizing hospital visits.(35) These findings underscore the need for further studies to refine cut-offs and optimize the use of HbA1c in GDM screening, as supported by our study's results.”
Reviewer 2 Report
Comments and Suggestions for Authors
I have reviewed the manuscript from Casellas and coauthors. In my opinion, this is a well-written and clear study which was devoted to comparative analysis of two screening methods for gestational diabetes diagnosis. I should note that one of the crucial advantages of this manuscript is absolutely clear understanding and clarification of potential limitations, which critically reduced the number of questions for this study. As with any clinical study, this study causes some questions about significance of showing diagnostic improvements on physiological level and for further patient's prognosis - and one of my questions was devoted to involved molecular mechanisms in the efficiency of alternative screening methods. I rephrase my review, if it is possible and applicable for the editorial team.
I have read and analyzed the manuscript from Casellas and coauthors. In my opinion the manuscript is devoted to an interesting theme, but the significance of this content remains some questions. However, the manuscript is well written and designed study with clear limitations, which can significantly reduced the number of potential questions. I have some debatable questions which can improve discussion of presented results.
1.Did authors compare the efficacy of two protocols?
2.Which molecular mechanisms can explain advances of alternative screening methods? Which is the physiological relevance of more accurate GDM diagnostics in remote consequences?
3.Which diagnostic methods will be preferable in different types of medical centers? Large emergency hospitals can prefer more cheaper and fast methods whereas more specialized hospitals can prefer more expensive but more accurate protocols.
Thus, I think that the manuscript can be considered for publication and has high potential impact, but only after major revision.
Author Response
We sincerely thank the reviewer for their valuable feedback and insightful comments on our manuscript. The constructive suggestions of the reviewer have contributed significantly to enhancing the clarity and rigour of our study. We sincerely appreciate recognising the reviewer’s efforts to address limitations and the relevance of the study's findings.
We have answered all the Reviewer’s comments as follows:
I have reviewed the manuscript from Casellas and coauthors. In my opinion, this is a well-written and clear study which was devoted to comparative analysis of two screening methods for gestational diabetes diagnosis. I should note that one of the crucial advantages of this manuscript is absolutely clear understanding and clarification of potential limitations, which critically reduced the number of questions for this study. As with any clinical study, this study causes some questions about significance of showing diagnostic improvements on physiological level and for further patient's prognosis - and one of my questions was devoted to involved molecular mechanisms in the efficiency of alternative screening methods. I rephrase my review, if it is possible and applicable for the editorial team. I have read and analyzed the manuscript from Casellas and coauthors. In my opinion the manuscript is devoted to an interesting theme, but the significance of this content remains some questions. However, the manuscript is well written and designed study with clear limitations, which can significantly reduced the number of potential questions. I have some debatable questions which can improve discussion of presented results.
We sincerely thank the reviewer for their thoughtful and constructive feedback on our manuscript. We are particularly grateful for the recognition of the manuscript’s strengths, including the clarity of the study design and the transparent acknowledgement of its limitations. The reviewer’s questions provide valuable insights and offer an excellent opportunity to enhance the discussion and further improve the manuscript.
We have answered all the Reviewer 2 comments as follows:
1.Did authors compare the efficacy of two protocols?
Yes, the manuscript directly compares the standard screening protocol and the alternative screening protocol implemented during the COVID-19 pandemic. The standard method diagnosed a higher percentage of GDM cases (10.8%) compared to the alternative method (6.9%), highlighting differences in sensitivity. Despite this, the adjusted analyses showed that the primary and secondary outcomes (e.g., large-for-gestational-age infants, delivery complications, and neonatal outcomes) were similar between groups. These results suggest that, although the alternative protocol was less sensitive, it was still effective in identifying clinically relevant cases, especially under the constraints of the pandemic.
2.Which molecular mechanisms can explain advances of alternative screening methods? Which is the physiological relevance of more accurate GDM diagnostics in remote consequences?
While the study does not explicitly investigate molecular mechanisms, it builds on the premise that HbA1c reflects long-term glycemic exposure, while plasma glucose provides acute glycemic data. This combination improves feasibility during a pandemic by reducing the reliance on time-sensitive fasting tests. Accurate GDM diagnosis is critical to managing maternal hyperglycemia, which is associated with reduced risks of macrosomia, preeclampsia, and adverse neonatal outcomes. Long-term, better glycemic control may lower the risk of metabolic disorders in the offspring and reduce maternal cardiovascular risks.
3.Which diagnostic methods will be preferable in different types of medical centers? Large emergency hospitals can prefer more cheaper and fast methods whereas more specialized hospitals can prefer more expensive but more accurate protocols. Thus, I think that the manuscript can be considered for publication and has high potential impact, but only after major revision.
The alternative method demonstrated cost-effectiveness and required fewer resources, making it well-suited for large emergency hospitals or resource-limited settings where simplicity and rapid implementation are essential. On the other hand, more specialized centers with adequate resources may prioritize the standard method, which has higher sensitivity and can identify milder cases of GDM. Tailoring the choice of screening method to the healthcare setting ensures optimal utilization of resources while maintaining diagnostic efficiency.
Round 2
Reviewer 2 Report
Comments and Suggestions for Authors
Manuscript can be accepted for publication
Author Response
We have added the reference recommended.